

# Sushi barcoding in the UK: another kettle of fish

Sara G. Vandamme[1,*], Andrew M. Griffiths[2,3,*], Sasha-Ann Taylor[1], Cristina Di Muri[1], Elizabeth A. Hankard[1], Jessica A. Towne[2], Mhairi Watson[2] and Stefano Mariani[1]

[1] School of Environment & Life Sciences, University of Salford, Greater Manchester, UK
[2] School of Biological Sciences, University of Bristol, Bristol, UK
[3] Biosciences, College of Environment & Life sciences, University of Exeter, UK
[*] These authors contributed equally to this work.

## ABSTRACT

Although the spread of sushi restaurants in the European Union and United States is a relatively new phenomenon, they have rapidly become among the most popular food services globally. Recent studies indicate that they can be associated with very high levels (>70%) of fish species substitution. Based on indications that the European seafood retail sector may currently be under better control than its North American counterpart, here we investigated levels of seafood labelling accuracy in sushi bars and restaurants across England. We used the COI barcoding gene to screen samples of tuna, eel, and a variety of other products characterised by less visually distinctive 'white flesh'. Moderate levels of substitution were found (10%), significantly lower than observed in North America, which lends support to the argument that public awareness, policy and governance of seafood labels is more effective in the European Union. Nevertheless, the results highlight that current labelling practice in UK restaurants lags behind the level of detail implemented in the retail sector, which hinders consumer choice, with potentially damaging economic, health and environmental consequences. Specifically, critically endangered species of tuna and eel continue to be sold without adequate information to consumers.

Corresponding authors
Sara G. Vandamme, vandamme-sara@hotmail.com
Stefano Mariani, s.mariani@salford.ac.uk

## INTRODUCTION

Seafood is a popular and healthy food choice and, therefore, one of the most commonly traded food commodities in the world (*FAO, 2014*). Regardless of the growing demand, studies on seafood mislabelling have identified that consumers are still too often given insufficient, confusing or misleading information about the seafood they purchase (*Warner et al., 2013*; *Pramod et al., 2014*; *Cawthorn et al., 2015*; *Di Pinto et al., 2015*). Due to increasingly complex supply chains, it is often unclear where and when seafood fraud is actually taking place, but restaurants and take-aways have been identified as the worst point of consumption for species substitution (*Jacquet & Pauly, 2008*; *Warner et al., 2013*; *Bénard-Capelle et al., 2015*). For example, large studies across North America illustrate that sushi venues have the highest level of mislabelling (74%–16%), followed by restaurants

(38%) and grocery stores (18%) (*Warner et al., 2013*; *Pramod et al., 2014*; *Khaksar et al., 2015*). Such findings suggest that, as restaurants often represent the end-point of these long and intricate supply chains, without needing to comply with the standardised labelling practices of the retail sector, they could be consistently associated with the highest levels of substitution.

Seafood fraud encompasses any illegal activity that misrepresents the fish being purchased. Although some mislabelling may result from unintended human errors in identifying fish or their origin, often it is driven by economic gain, where cheaper or more readily available species are sold instead of expensive, desirable or supply-limited species e.g., farmed tilapia, *Oreochromis sp.*, sold as snapper, *Lutjanus sp.*, (*Jacquet & Pauly, 2008*; *Warner et al., 2013*). Mislabelling can also provide cover and profit for illegal and unregulated fishing and seafood (*Watson et al., 2015*), which could have damaging implications for fisheries management and conservation, e.g., Atlantic halibut *Hippoglossus hippoglossus* sold as Pacific halibut *Hippoglossus stenolepis* (*Warner et al., 2013*). Seafood fraud can also have serious health consequences when mislabelled seafood masks undeclared allergens, contaminants or toxins. This is exemplified by escolar, *Lepidocybium flavobrunneum*, sold as "white tuna" (*Lowenstein, Burger & Jeitner, 2010*; *Warner et al., 2013*); escolar can naturally contain a toxin, gempylotoxin, which can cause mild to severe gastrointestinal problems, meaning this species is banned from the market in Italy and Japan.

The European Union (EU) is the largest single market for imported fish and fishery products, representing about 23% of world imports, and continuing to grow (*FAO, 2014*). As such, the EU has a great responsibility to demonstrate legal and sustainable seafood supply chains to consumers. Its illegal fishing regulation (EC No 1005/2008) is an innovative and pioneering legal tool that has placed the EU at the forefront of global efforts to address illegal, unreported and unregulated (IUU) fishing. Part of the ongoing legal framework is the new European regulation (EC No 1379/2013), enacted in December 2014, which places an onus on anybody selling seafood to label it clearly and accurately, providing consumers with highly transparent information. This new EU labelling legislation applies to all pre-packed and non-packed fishery and aquaculture products (excluding preserved and prepared meals) at all stages in the retail supply chain, but excludes restaurants, which only have to provide mandatory information on allergens. In other words, restaurants are not obliged to mention on their menu what species is being sold but they are obliged to keep and give this information to the consumer if asked for. Additionally, EU Member States have to draw up a list of the commercial designations accepted in their territory, together with their scientific names. However, for some groups, like eels or tunas, the authorized commercial names cover a large number of species, including those with serious conservation concern. In such cases, there is no way for knowledgeable consumers to choose according to sustainability criteria.

Given recent indication that the European seafood retail sector may have significantly lower levels of fraudulent substitutions than its North American counterpart (*Bénard-Capelle et al., 2015*; *Helyar et al., 2014*; *Mariani et al., 2015*), we set out to investigate the levels of seafood mislabelling in Britain's raw seafood restaurants. Since sushi venues were so susceptible to fraud in the American seafood trade (*Lowenstein, Amato & Kolokotronis,*

*2009*; *Warner et al., 2013*), we focussed on this specific part of the supply chain. Sampling was spread across six different cities, focussing on tuna, eel and opportunistic samples of less distinguishable white-fleshed fish.

## MATERIALS AND METHODS

### Sampling

A total of 115 fish samples were collected in 31 sushi restaurants in Manchester, London, Bristol, Liverpool, Exeter and Newcastle, between September 2014 and 2015. Two independent sets of samples were collected in restaurants in Manchester, Liverpool, and Newcastle, with a minimum of two weeks between sampling. In all cases the individuals involved in the collection of tissue posed as normal customers and sampled in an as unobtrusive way as possible.

Samples were placed in pre-numbered tubes and stored in 95% ethanol at $-20\,°C$ until extraction. Data were recorded, including commercial name, date, price, location, restaurant name, as well as photographs of samples when possible. Sampling focused on tuna (*Thunnus* sp.) and eel (*Anguilla* sp.) samples; these two product types are highly sought-after and include critically endangered species. A selection of less distinguishable white-fleshed fish available in each restaurant was also collected (Table 1) as these can comprise hundreds of fish species whose flesh is virtually unrecognisable by consumers and hence easily susceptible to substitution.

### DNA extraction and sequencing

Genomic DNA was extracted from muscle tissue according to a Chelex resin protocol (*Estoup et al., 1996*). The partial cytochromoxidase 1 (COI) was amplified using the FishF2 and FishR2 from *Ward et al. (2005)*, following the PCR amplifications by *Serra-Pereira et al. (2010)*. If samples could not be successfully amplified, the COI mini-barcode primers (mICOIintF and jgHCO2198) following *Leray et al. (2013)* or the L14735 and H15149 cytochrome b (cytb) primers as described by *Burgener (1997)* were used. In the case of cytb amplification, 2 µl 10× reaction buffer, 1.6 µl MgCl2 (50 mM), 1 µl of each primer (0.01 mM), 0.5 Units of DNA Taq Polymerase (PROMEGA, Madison, WI, USA) and 0.2 µl of each dNTP (10 µM) were used in a total volume of 20 µL. PCR conditions entailed 5 min at 94 °C, following a cycle of 40 s at 94 °C, 80 s at 55 °C, 80 s at 72 °C, which is repeated 35 times, finalized by 7 min at 72 °C, until the PCR was held at 10 °C.

DNA sequencing was carried out by Source Bioscience (Cambridge, UK) and all sequences were obtained with the forward primer. Sequences were checked manually against their chromatogram and edited in BioEdit (*Hall, 1999*). Each sequence was then used to BLAST-search both the GenBank reference database (www.ncbi.nlm.nih.gov/) and the Barcode of Life Data system (BOLD, http://www.boldsystems.org/, see *Ratnasingham & Hebert, 2007*), using the "Public Record Barcode Database", which restricts the search to sequences that have been published. In the Supplemental Information, results are presented for the alternative BOLD reference databases: the default "Species Level Barcode Records" database and the "Full Length Record Barcode Database", which is recommended to use with short sequences as it provides a maximum overlap. Identification was determined

Vandamme et al. (2016), *PeerJ*, DOI 10.7717/peerj.1891

Peer J

**Table 1  Summary of the samples collected in sushi venues across the UK.** Identification represented in this table is obtained by using the BOLD 'Public Record Barcode' database. Samples marked by (∗) represent samples which were identified using cyt *b* sequencing and the Genbank public database, the (a) characterises samples identified by the COI mini-barcodes. Results by using other database can be found in Table S1. The conservation status of the species can by assessed by their IUCN Red List of Threatened Species status.

| City | Sold as | BOLD Public Record Barcode database (% match) | Accepted common name | Mislabelled | IUCN status | Accession number |
|---|---|---|---|---|---|---|
| Bristol | Tuna (Albacore) | *Thunnus alalunga* 100%, *Thunnus obesus* 100%, *Thunnus orientalis* 99.81%, *Thunnus thynnus* 99.61%, *Thunnus atlanticus* 99.03% | Albacore | NO | Near threatened | KU168615 |
| Exeter | Tuna (Albacore) | *Thunnus alalunga* 100%, *Thunnus obesus* 100%, *Thunnus orientalis* 99.81%, *Thunnus maccoyii* 99.81%, *Thunnus atlanticus* 99.04% | Albacore | NO | Near threatened | KU168616 |
| London | Tuna (Albacore) | *Thunnus alalunga* 99.79%, *Thunnus obesus* 99.38%, *Thunnus orientalis* 99.17%, *Thunnus maccoyii* 99.17%, *Thunnus thynnus* 98.96%, *Thunnus albacares* 98.33% | Albacore | NO | Near threatened | KU168617 |
| Bristol | Tuna (Bluefin) | *Thunnus thynnus* 100% | Atlantic Bluefin tuna | NO | Endangered | KU168618 |
| Liverpool | Tuna (Bluefin) | *Thunnus albacares* 100%, *Thunnus atlanticus* 100%, *Thunnus obesus* 100%, *Thunnus maccoyii* 99.85% | Yellowfin tuna | YES | Near threatened | KU168619 |
| Bristol | Tuna (Yellowfin) | *Thunnus albacares* 100%, *Thunnus obesus* 100%, *Thunnus maccoyii* 99.83%, *Thunnus tonggol* 99.83% | Yellowfin tuna | NO | Near threatened | KU168620 |
| Bristol | Tuna (Yellowfin) | *Thunnus albacares* 100%, *Thunnus atlanticus* 100%, *Thunnus obesus* 100%, *Thunnus maccoyii* 99.83%, *Thunnus tonggol* 99.83% | Yellowfin tuna | NO | Near threatened | KU168621 |
| Exeter | Tuna (Yellowfin) | *Thunnus albacares* 100%, | Yellowfin tuna | NO | Near threatened | KU168622 |
| London | Tuna (Yellowfin) | *Thunnus albacares* 100%, *Thunnus atlanticus* 99.79%, *Thunnus obesus* 99.79%, *Thunnus maccoyii* 99.79% | Yellowfin tuna | NO | Near threatened | KU168623 |

Vandamme et al. (2016), *PeerJ*, DOI 10.7717/peerj.1891

**Table 1** (*continued*)

| City | Sold as | BOLD Public Record Barcode database (% match) | Accepted common name | Mislabelled | IUCN status | Accession number |
|------|---------|----------------------------------------------|---------------------|-------------|-------------|------------------|
| Manchester | Tuna (Yellowfin) | *Thunnus obesus* 100%, *Thunnus albacares* 99.69%, *Thunnus atlanticus* 99.62%, *Thunnus tonggol* 99.52%, *Thunnus maccoyii* 99.4% | Bigeye tuna | YES | Vulnerable | KU168624 |
| Manchester | Tuna (Yellowfin) | *Thunnus albacares* 100%, *Thunnus atlanticus* 100%, *Thunnus maccoyii* 100%, *Thunnus obesus* 100% | Yellowfin tuna | NO | Near threatened | KU168625 |
| Bristol | Tuna | *Thunnus albacares* 100%, *Thunnus obesus* 99.82%, *Thunnus maccoyii* 99.67%, *Thunnus tonggol* 99.67% | Yellowfin tuna | NO | Near threatened | KU168627 |
| Bristol | Tuna | *Thunnus albacares* 100%, *Thunnus obesus* 99.82%, *Thunnus maccoyii* 99.67%, *Thunnus tonggol* 99.67% | Yellowfin tuna | NO | Near threatened | KU168628 |
| Bristol | Tuna | *Thunnus albacares* 100%, *Thunnus atlanticus* 100%, *Thunnus obesus* 100%, *Thunnus maccoyii* 99.83%, *Thunnus tonggol* 99.83% | Yellowfin tuna | NO | Near threatened | KU168629 |
| Bristol | Tuna | *Thunnus albacares* 100%, *Thunnus atlanticus* 100%, *Thunnus obesus* 100%, *Thunnus maccoyii* 99.83%, *Thunnus tonggol* 99.83% | Yellowfin tuna | NO | Near threatened | KU168630 |
| Bristol | Tuna | *Thunnus obesus* 100%, *Thunnus albacares* 99.34% | Bigeye tuna | NO | Vulnerable | KU168631 |
| Bristol | Tuna | *Thunnus albacares* 100%, *Thunnus obesus* 99.83% | Yellowfin tuna | NO | Near threatened | KU168632 |
| Bristol | Tuna | *Thunnus albacares* 100%, *Thunnus atlanticus* 100%, *Thunnus maccoyii* 100%, *Thunnus obesus* 100%, *Thunnus tonggol* 99.84% | Yellowfin tuna | NO | Near threatened | KU168633 |
| Bristol | Tuna | *Thunnus albacares* 100%, *Thunnus atlanticus* 100%, *Thunnus obesus* 100%, *Thunnus maccoyii* 99.84%, *Thunnus tonggol* 99.83% | Yellowfin tuna | NO | Near threatened | KU168634 |

Vandamme et al. (2016), *PeerJ*, DOI 10.7717/peerj.1891

| City | Sold as | BOLD Public Record Barcode database (% match) | Accepted common name | Mislabelled | IUCN status | Accession number |
|---|---|---|---|---|---|---|
| Exeter | Tuna | *Thunnus albacares* 100%, *Thunnus atlanticus* 99.49%, *Thunnus obesus* 99.49%, *Thunnus maccoyii* 99.48% | Yellowfin tuna | NO | Near threatened | KU168635 |
| Liverpool | Tuna* | *Thunnus albacares* 100%, *Thunnus obesus* 100%, *Thunnus atlanticus* 99.81%, *Thunnus maccoyii* 99.68% | Yellowfin tuna | NO | Near threatened | KU168636 |
| Liverpool | Tuna | *Thunnus albacares* 100% | Yellowfin tuna | NO | Near threatened | KU168637 |
| London | Tuna | *Thunnus albacares* 100%, *Thunnus maccoyii* 100%, *Thunnus obesus* 100% | Yellowfin tuna | NO | Near threatened | KU168638 |
| London | Tuna | *Thunnus albacares* 100%, *Thunnus atlanticus* 100%, *Thunnus maccoyii* 100%, *Thunnus obesus* 100% | Yellowfin tuna | NO | Near threatened | KU168639 |
| London | Tuna | *Seriola lalandi* 100%, *Seriola zonata* 99.36% | Yellowtail amberjack | YES | Not assessed | KU168640 |
| London | Tuna | *Seriola lalandi* 100%, *Seriola zonata* 99.36% | Yellowtail amberjack | YES | Not assessed | KU168641 |
| London | Tuna | *Thunnus albacares* 99.82%, *Thunnus atlanticus* 99.82%, *Thunnus maccoyii* 99.82%, *Thunnus obesus* 99.81% | Yellowfin tuna | NO | Near threatened | KU168642 |
| London | Tuna | *Thunnus albacares* 100%, *Thunnus atlanticus* 99.79%, *Thunnus obesus* 99.79%, *Thunnus maccoyii* 99.79% | Yellowfin tuna | NO | Near threatened | KU168643 |
| London | Tuna | *Thunnus albacares* 99.79%, *Thunnus atlanticus* 99.79%, *Thunnus obesus* 99.79%, *Thunnus maccoyii* 99.79% | Yellowfin tuna | NO | Near threatened | KU168644 |
| London | Tuna | *Thunnus albacares* 100%, *Thunnus atlanticus* 100%, *Thunnus maccoyii* 100%, *Thunnus obesus* 100% | Yellowfin tuna | NO | Near threatened | KU168645 |
| London | Tuna[a] | *Thunnus thynnus* 100% | Atlantic Bluefin tuna | NO | Endangered | KU168646 |
| London | Tuna | *Thunnus albacares* 100%, *Thunnus atlanticus* 100%, *Thunnus maccoyii* 100%, *Thunnus obesus* 100% | Yellowfin tuna | NO | Near threatened | KU168647 |

Vandamme et al. (2016), *PeerJ*, DOI 10.7717/peerj.1891

**Table 1** (*continued*)

| City | Sold as | BOLD Public Record Barcode database (% match) | Accepted common name | Mislabelled | IUCN status | Accession number |
|------|---------|-----------------------------------------------|----------------------|-------------|-------------|------------------|
| London | Tuna | *Thunnus albacares* 100% | Yellowfin tuna | NO | Near threatened | KU168648 |
| London | Tuna | *Thunnus albacares* 100%, *Thunnus atlanticus* 100%, *Thunnus maccoyii* 100%, *Thunnus obesus* 100% | Yellowfin tuna | NO | Near threatened | KU168649 |
| Manchester | Tuna* | *Seriola quinqueradiata* 99.85%, *Seriola lalandi* 94.97% | Japanese amberjack | YES | Not assessed | KU168650 |
| Manchester | Tuna | *Thunnus obesus* 100%, *Thunnus albacares* 99.69% | Bigeye tuna | NO | Vulnerable | KU168651 |
| Manchester | Tuna* | *Thunnus albacares* 100%, *Thunnus atlanticus* 100%, *Thunnus obesus* 100%, *Thunnus maccoyii* 99.85% | Yellowfin tuna | NO | Near threatened | KU168652 |
| Manchester | Tuna (Spicy) | *Thunnus albacares* 100%, *Thunnus atlanticus* 100%, *Thunnus obesus* 100%, *Thunnus maccoyii* 99.85% | Yellowfin tuna | NO | Near threatened | KU168653 |
| Manchester | Tuna | *Thunnus albacares* 100%, *Thunnus atlanticus* 100%, *Thunnus obesus* 100%, *Thunnus maccoyii* 99.85% | Yellowfin tuna | NO | Near threatened | KU168654 |
| Manchester | Tuna | *Thunnus thynnus* 100%, *Thunnus orientalis* 99.69%, *Thunnus atlanticus* 99.69%, *Thunnus maccoyii* 99.54%, *Thunnus albacares* 99.53% | Atlantic Bluefin tuna | NO | Endangered | KU168655 |
| Manchester | Tuna | *Thunnus albacares* 100% | Bigeye tuna | NO | Vulnerable | KU168656 |
| Manchester | Tuna | *Thunnus albacares* 100% | Yellowfin tuna | NO | Near threatened | KU168657 |
| Manchester | Tuna | *Thunnus thynnus* 100%, *Thunnus orientalis* 99.84%, *Thunnus maccoyii* 99.84%, *Thunnus alalunga* 99.69%, *Thunnus obesus* 99.68%, *Thunnus atlanticus* 99.19%, *Thunnus albacares* 99% | Atlantic Bluefin tuna | NO | Endangered | KU168658 |
| Manchester | Tuna | *Thunnus albacares* 100%, *Thunnus maccoyii* 99.84%, *Thunnus obesus* 99.82%, *Thunnus atlanticus* 99.8% | Yellowfin tuna | NO | Near threatened | KU168659 |

Vandamme et al. (2016), *PeerJ*, DOI 10.7717/peerj.1891

Peer J

| City | Sold as | BOLD Public Record Barcode database (% match) | Accepted common name | Mislabelled | IUCN status | Accession number |
|------|---------|-----------------------------------------------|----------------------|-------------|-------------|------------------|
| Manchester | Tuna* | *Thunnus albacares* 100%, *Thunnus atlanticus* 100%, *Thunnus maccoyii* 100%, *Thunnus obesus* 100%, *Thunnus tonggol* 99.84% | Yellowfin tuna | NO | Near threatened | KU168660 |
| Manchester | Tuna | *Thunnus albacares* 100%, *Thunnus atlanticus* 100%, *Thunnus maccoyii* 100%, *Thunnus obesus* 100% | Yellowfin tuna | NO | Near threatened | KU168661 |
| Newcastle | Tuna | *Thunnus albacares* 100%, *Thunnus atlanticus* 100%, *Thunnus obesus* 100%, *Thunnus maccoyii* 99.85%, | Yellowfin tuna | NO | Near threatened | KU168662 |
| Newcastle | Tuna | *Thunnus albacares* 100%, *Thunnus atlanticus* 99.84%, *Thunnus maccoyii* 99.84%, *Thunnus obesus* 99.82% | Yellowfin tuna | NO | Near threatened | KU168663 |
| Bristol | Eel | *Anguilla anguilla* 100% | European eel | NO | Critically endangered | KU168664 |
| Bristol | Eel | *Anguilla anguilla* 100% | European eel | NO | Critically endangered | KU168665 |
| Bristol | Eel | *Anguilla marmorata* 99.84% | Giant mottled eel | NO | Least concern | KU168666 |
| Exeter | Eel | *Anguilla japonica* 99.36% | Japanese eel | NO | Endangered | KU168667 |
| Liverpool | Eel | *Anguilla anguilla* 99.84% | European eel | NO | Critically endangered | KU168668 |
| Liverpool | Eel | *Anguilla rostrata* 99.84% | American eel | NO | Endangered | KU168669 |
| Liverpool | Eel | *Anguilla japonica* 100% | Japanese eel | NO | Endangere d | KU168670 |
| London | Eel (Freshwater)[a] | *Anguilla japonica* 100% | Japanese eel | NO | Endangered | KU168671 |
| London | Eel (grilled) | *Anguilla anguilla* 100% | European eel | NO | Critically endangered | KU168672 |
| London | Eel[a] | *Anguilla japonica* 99.49% | Japanese eel | NO | Endangered | KU168673 |
| Manchester | Eel | *Anguilla anguilla* 99.84% | European eel | NO | Critically endangered | KU168674 |
| Manchester | Eel (Freshwater) | *Anguilla anguilla* 100% | European eel | NO | Critically endangered | KU168675 |
| Manchester | Eel | *Anguilla anguilla* 99.84% | European eel | NO | Critically endangered | KU168676 |
| Manchester | Eel | *Anguilla japonica* 99.54%, *Anguilla marmorata* 94.74% | Japanese eel | NO | Endangered | KU168677 |

Vandamme et al. (2016), *PeerJ*, DOI 10.7717/peerj.1891

**Table 1** (*continued*)

| City | Sold as | BOLD Public Record Barcode database (% match) | Accepted common name | Mislabelled | IUCN status | Accession number |
|------|---------|-----------------------------------------------|----------------------|-------------|-------------|------------------|
| Manchester | Eel | *Anguilla rostrata* 99.84% | American eel | NO | Endangered | KU168678 |
| Manchester | Eel | *Anguilla anguilla* 100% | European eel | NO | Critically endangered | KU168679 |
| Manchester | Eel | *Anguilla anguilla* 100% | European eel | NO | Critically endangered | KU168680 |
| Manchester | *Eel*\* | *Anguilla anguilla* 90% | European eel | NO | Critically endangered | KU168681 |
| Newcastle | Eel | *Anguilla anguilla* 100% | European eel | NO | Critically endangered | KU168683 |
| Newcastle | Eel | *Anguilla anguilla* 99.37% | European eel | NO | Critically endangered | KU168684 |
| Liverpool | *Seabass*\* | *Dicentrarchus labrax* 99% | European seabass | NO | Least concern | KU168685 |
| Liverpool | *Seabass*\* | *Dicentrarchus labrax* 100% | European seabass | NO | Least concern | KU168686 |
| Liverpool | Seabass[a] | *Dicentrarchus labrax* 100% | European seabass | NO | Least concern | KU168687 |
| London | *Seabass*\* | *Dicentrarchus labrax* 100% | European seabass | NO | Least concern | KU168688 |
| London | Seabass | *Lateolabrax japonicus* 100%, *Lateolabrax maculatus* 99.63% | Japanese seabass | YES | Not assessed | KU168689 |
| London | Seabass | *Lateolabrax japonicus* 100%, *Lateolabrax maculatus* 99.49% | Japanese seabass | YES | Not assessed | KU168690 |
| London | *Seabass*\* | *Dicentrarchus labrax* 100% | European seabass | NO | Least concern | KU168691 |
| London | *Seabass*\* | *Dicentrarchus labrax* 100% | European seabass | NO | Least concern | KU168692 |
| London | *Seabass*\* | *Dicentrarchus labrax* 100% | European seabass | NO | Least concern | KU168693 |
| Manchester | *Seabass*\* | *Dicentrarchus labrax* 99% | European seabass | NO | Least concern | KU168694 |
| Manchester | *Seabass*\* | *Dicentrarchus labrax* 99% | European seabass | NO | Least concern | KU168695 |
| Manchester | *Seabass*\* | *Dicentrarchus labrax* 99% | European seabass | NO | Least concern | KU168696 |
| Manchester | *Seabass*\* | *Dicentrarchus labrax* 100% | European seabass | NO | Least concern | KU168697 |
| Manchester | *Seabass*\* | *Dicentrarchus labrax* 100% | European seabass | NO | Least concern | KU168698 |
| Manchester | *Seabass*\* | *Dicentrarchus labrax* 100% | European seabass | NO | Least concern | KU168699 |
| Manchester | *Seabass*\* | *Dicentrarchus labrax* 100% | European seabass | NO | Least concern | KU168700 |

Vandamme et al. (2016), *PeerJ*, DOI 10.7717/peerj.1891

**Table 1** (*continued*)

| City | Sold as | BOLD Public Record Barcode database (% match) | Accepted common name | Mislabelled | IUCN status | Accession number |
|------|---------|-----------------------------------------------|----------------------|-------------|-------------|------------------|
| Bristol | Yellowtail | *Seriola quinqueradiata* 99.34%, *Seriola lalandi* 94.53% | Japanese amberjack | NO | Not assessed | KU168701 |
| Bristol | Yellowtail | *Seriola quinqueradiata* 99.51%, *Seriola lalandi* 94.75% | Japanese amberjack | NO | Not assessed | KU168702 |
| Bristol | Yellowtail | *Seriola quinqueradiata* 99.84%, *Seriola lalandi* 94.9% | Japanese amberjack | NO | Not assessed | KU168703 |
| Liverpool | Yellowtail | *Seriola quinqueradiata* 99.63%, *Seriola lalandi* 93.85% | Japanese amberjack | NO | Not assessed | KU168704 |
| London | Yellowtail | *Seriola quinqueradiata* 99.69% | Japanese amberjack | NO | Not assessed | KU168705 |
| London | Yellowtail | *Seriola lalandi* 100%, *Seriola zonata* 99.34% | Yellowtail amberjack | NO | Not assessed | KU168706 |
| London | Yellowtail | *Seriola quinqueradiata* 99.80% | Japanese amberjack | NO | Not assessed | KU168707 |
| London | Yellowtail | *Seriola quinqueradiata* 99.79% | Japanese amberjack | NO | Not assessed | KU168708 |
| London | Yellowtail | *Seriola quinqueradiata* 99.79% | Japanese amberjack | NO | Not assessed | KU168709 |
| London | Yellowtail | *Seriola quinqueradiata* 99.77% | Japanese amberjack | NO | Not assessed | KU168710 |
| Manchester | Yellowtail | *Seriola quinqueradiata* 99.55%, *Seriola lalandi* 94.97% | Japanese amberjack | NO | Not assessed | KU168711 |
| Manchester | Yellowtail | *Seriola quinqueradiata* 99.7%, *Seriola lalandi* 94.9% | Japanese amberjack | NO | Not assessed | KU168712 |
| London | Mackerel | *Scomber scombrus* 100% | Mackerel | NO | Least concern | KU168713 |
| London | Mackerel | *Scomber scombrus* 99.80% | Mackerel | NO | Least concern | KU168714 |
| London | Mackerel | *Scomber scombrus* 100% | Mackerel | NO | Least concern | KU168715 |
| London | Mackerel | *Scomber scombrus* 100% | Mackerel | NO | Least concern | KU168716 |
| London | Mackerel | *Scomber scombrus* 100% | Mackerel | NO | Least concern | KU168717 |
| London | Mackerel | *Scomber scombrus* 100% | Mackerel | NO | Least concern | KU168718 |
| London | Mackerel | *Scomber scombrus* 100% | Mackerel | NO | Least concern | KU168719 |
| London | Mackerel | *Scomber scombrus* 100% | Mackerel | NO | Least concern | KU168720 |
| Manchester | Seabream | *Sparus aurata* 100% | Gilthead bream | NO | Least concern | KU168721 |
| Manchester | Seabream | *Sparus aurata* 100% | Gilthead bream | NO | Least concern | KU168722 |
| Manchester | Seabream | *Sparus aurata* 100% | Gilthead bream | NO | Least concern | KU168723 |
| Liverpool | Swordfish | *Makaira nigricans* 99.52% | Blue marlin | YES | Data deficient | KU168724 |
| Newcastle | Swordfish | *Xiphias gladius* 100% | Swordfish | NO | Least concern | KU168725 |

Vandamme et al. (2016), *PeerJ*, DOI 10.7717/peerj.1891

**Table 1** (*continued*)

| City | Sold as | BOLD Public Record Barcode database (% match) | Accepted common name | Mislabelled | IUCN status | Accession number |
|------|---------|----------------------------------------------|---------------------|-------------|-------------|------------------|
| London | King Fish | *Seriola lalandi* 100%, *Seriola zonata* 99.38% | Yellowtail amberjack | YES | Not assessed | KU168726 |
| Manchester | King Fish (Tasmanian) | *Seriola lalandi* 100%, *Seriola zonata* 99.43% | Yellowtail amberjack | YES | Not assessed | KU168727 |
| Manchester | Barramundi[a] | *Lates calcarifer* 100% | Barramundi | NO | Not assessed | KU168728 |
| Manchester | Black Cod | *Anoplopoma fimbria* 100% | Sablefish | NO | Not assessed | KU168729 |
| Liverpool | Flying Fish eggs | *Clupea harengus* 100% | Herring | YES | Least concern | KU168731 |
| London | Snapper | *Sparus aurata 100%* | Gilthead bream | YES | Least concern | KU168732 |

by sequence similarity to the reference dataset (*Wong & Hanner, 2008*), and checked by "Tree based identification" (i.e., distance trees in BOLD; *Costa et al., 2012*). With the NCBI database a minimum similarity of 90% was required. The match with the highest expectation value (*E*-value) of the BLAST program was retained as potential species identification. The *E*-value is a parameter that describes the number of hits one can expect to see just by chance when searching a database of a particular size.

For each sample, the list of admissible species that can be sold under the commercial name indicated on the menu was determined by consulting the UK governmental list with commercial designations of fish (*DEFRA, 2013*). The sample was declared mislabelled if the species name determined through molecular identification did not match the commercially accepted names in this list. Species or commercial names obtained orally from waiting staff in restaurants were not utilised in calculations of substitution rates, but this information is available in Table S1.

## RESULTS AND DISCUSSION

This study represents the largest sampling of UK sushi venues to date. A relatively intensive effort was made to collect samples across multiple time-points and regions, going beyond the sampling of only the most commonly consumed species like tuna, eel and salmon. The inherently high cost of sampling raw fish restaurants as consumers represents a limitation to the collection of huge sample sizes. However, the final sample size ($N = 115$) is of the same order of magnitude as recent comparable investigations and the sample design that was spread over 31 restaurants and a 12-month span, strove to avoid high levels of repeated sampling from any one location or restaurant, giving a degree of independence to the data.

Interpretable sequences were obtained for a total of 115 samples, ranging between 166 and 674 base pairs (bp) (average length 531 bp). These include 48 'tuna', 20 'eel', 16 'seabass', 12 'yellowtail', 8 'mackerel', 3 'seabream', 2 'swordfish', 2 'kingfish', and single samples of 'black cod', 'barramundi', 'snapper' and 'flying fish' (Table 1). Searches on BOLD and GenBank generally produced clear matches allowing for confident assignment of species and there was good agreement between databases (Table S1). In fact, all searches yielded matches that were within the 98% similarity to database records. For all sea bass samples and one eel sample, no successful COI amplifications could be produced, and the cytb primers were utilised instead. A BOLD search could not be made in these instances, as this database only contains COI sequences, so the GenBank identification was used.

In the case of certain *Thunnus* species, little interspecific divergence can limit the power of COI to discriminate among species pair, owing to the short evolutionary history and/or introgression among them (*Tseng et al., 2012*; *Vinas & Tudela, 2009*). However, in the current study this would not generally cause issues in assessing the levels of substitution as the commercial designation by *DEFRA (2013)* allows restaurants to sell all *Thunnus* species under the umbrella term "tuna" . Despite the limitation in *Thunnus* identification, in some instances there is the potential to go down to species level identification. We can distinguish *T. thynnus* from the other *Thunnus* species by following a set of criteria. First, when there is 100% sequence match criterion alongside the reduced similarity between

the unknown sequences and any other matching species record. Second, the phylogenetic tree option in the BOLD reference database provides further evidence of the origin of the species. Finally, comparison of results of different/more stringent sets of reference data in BOLD further provides an unambiguous identification. Therefore, it was possible with some samples to assign the sequence obtained to either the yellowfin or bluefin tuna group, providing evidence of mislabelling.

The overall level of mislabelling and substitution was moderate (10.4%, Table 2). In the case of tuna, three samples were sold as tuna, but identified as Yellowtail and Japanese Amberjack (*Seriola lalandi* and *Seriola quinqueradiata*, respectively). In two other cases, the restaurant deliberately advertised a specific *Thunnus* species: one restaurant claimed to sell Yellowfin tuna (*Thunnus albacares*) while highest similarity scores by COI barcoding suggested potential substitution with Big-eye tuna (*Thunnus obesus*). Another restaurant claimed to serve Bluefin tuna, but COI barcoding revealed matches with Big-eye and Yellowfin tuna. Although the common name Bluefin tuna encompasses Atlantic Bluefin (*Thunnus thynnus*), Pacific Bluefin (*Thunnus orientalis*) and Southern Bluefin (*Thunnus maccoyii*), none of them matched the COI barcoding results. Kingfish was sampled in London and Manchester. According to the official list on commercial designation of fish in the United Kingdom (DEFRA, 2013) this common name represents all species of *Scomberomorus.* However, both samples were identified as *Seriola lalandi* and hence regarded as mislabelled. Among the 16 samples of seabass, two samples were identified as *Lateolabrax maculaus* also known as the Japanese seabass. In the case of one "swordfish" sample, the reference database inquiry identified the species *Makaira nigricans* (Atlantic blue marlin), with additional matches from closely related sister taxa belonging to other marlin species (Family: Istiophoridae). Although it is difficult to pinpoint the exact species ID, it is evident that the sample did not match with swordfish (*Xiphias gladius).* Further mislabelling was found for a sample of snapper (Family: Lutjanidae) which was identified as *Sparus aurata* (gilt-head sea bream) and the sample of the flying fish eggs (representing all species of the family Exocoetidae) were identified as herring (*Clupea harengus*) eggs. The sample of Black cod was identified as *Anoplopoma fimbria*. According to Fishbase, both Black cod and Sablefish are accepted common names for *Anoplopoma fimbria;* however, the official list on commercial designation of fish in the United Kingdom (DEFRA, 2013) only accepts 'sablefish'. As both common names are accepted by the scientific community, this particular example was not deemed to be mislabelled, as the restaurant business aimed to serve a rather unfamiliar species to the UK public and used a scientifically correct name. Rather than mislabelling, this example can be seen as a misapplied market nomenclature, which shows how, in a context of increasingly global and diverse seafood market, regular communication between governments, fisheries managers and scientific advisors should be improved in order to guarantee an updated and accurate list of valid names. Yet, the new labelling regulations (EC 1379/2013, article 37) requiring the use of scientific names, may offer the necessary level of universality to commercial designations.

When compared to recent studies on sushi labelling in North America, which returned 74% (Warner et al., 2013) and 16.3% (Khaksar et al., 2015) in the level of substitution, the UK food service sector comes under a more positive light (Table 1 and Fig. 1). Similarly,

Vandamme et al. (2016), *PeerJ*, DOI 10.7717/peerj.1891

**Table 2  Samples collected across the UK per species and per city.**

| City | "Tuna" | "Eel" | Seabass | Yellowtail | Seabream | Mackerel | Swordfish | Black cod | Barramundi | Kingfish | Snapper | Flying fish eggs | TOTAL |
|---|---|---|---|---|---|---|---|---|---|---|---|---|---|
| *Manchester* | 14 | 8 | 7 | 2 | 3 | | | 1 | 1 | 1 | | | 37 |
| *London* | 14 | 3 | 6 | 6 | | 8 | | | | 1 | 1 | | 39 |
| *Bristol* | 12 | 3 | | 3 | | | | | | | | | 18 |
| *Liverpool* | 3 | 3 | 3 | 1 | | | 1 | | | | | 1 | 12 |
| *Newcastle* | 2 | 2 | | | | | 1 | | | | | | 5 |
| *Exeter* | 3 | 1 | | | | | | | | | | | 4 |
| TOTAL mislabelled | 5 | 0 | 2 | 0 | 0 | 0 | 1 | 0 | 0 | 2 | 1 | 1 | 12 |
| TOTAL | 48 | 20 | 16 | 12 | 3 | 8 | 2 | 1 | 1 | 2 | 1 | 1 | **115** |
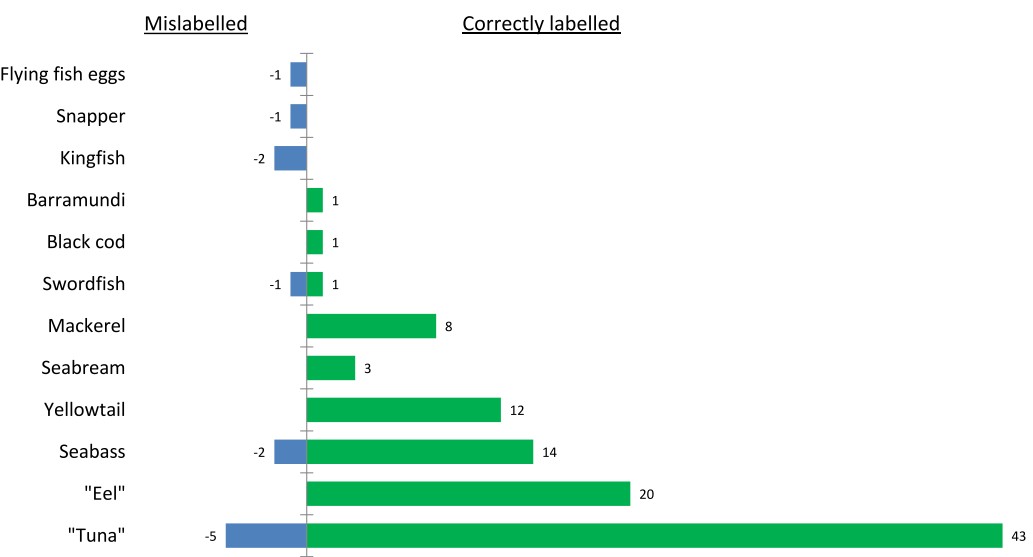

**Figure 1  Level of mislabelling per species.** For the two 'Swordfish' samples, one sample was found correctly labelled, where the other was substituted with Marlin. Both the Marlin and Swordfish are depicted on either side of the diagram. Furthermore, substitution was recorded in tuna, seabass, kingfish, snapper and flying fish eggs samples.

*Bénard-Capelle et al. (2015)* found only 3% substitution in French restaurants, which suggests lower levels of mislabelling in restaurants across Europe. In contrast to North America, mislabelling of tuna is less pronounced (10.2%). Generally in Europe substitution occurred between tuna species (*Bénard-Capelle et al., 2015*), or with amberjack, unlike in the US where a large portion of the tuna is substituted with escolar (*Lepidocybium flavobrunneum, Warner et al., 2013*). Comparisons between mislabelling in North America and the EU are valid as labelling regulation for the *FDA (2016)* and the EU are similar as to allowing umbrella term to be used for the sale of product in restaurants. Interestingly, in one case where oral enquiry about which tuna species was being sold was made to the waiting staff, the response was Bluefin tuna, which was not supported by the results of DNA barcoding. In this study, it was not included as a case of mislabelling, as the menu did not explicitly mention "Bluefin tuna", but it does illustrate an absence of care or knowledge in the usage of this commercial name. Given that consumers are not expected to know every possible regional name, and the need to standardise labels across a large region with many different languages, the EU's policy to require scientific names on display appears inevitable. The lowest level of mislabelling among the most studies detected only 16.3% of mislabelling in North America (*Khaksar et al., 2015*). In spite of the short sampling time and moderate samples size, their result is in sharp contrast to the study by *Warner et al. (2013)* who detected 74% mislabelling, suggesting a decreasing trends in mislabelling and illustrating that the role of media, environmental Non-governmental Organisations and scientific outputs in increasing public awareness is undeniable, which in turn raises the demand for enforcement of more rigorous inspection and audit processes in the food supply chain. Surveillance studies like this can help further refine the scope of such efforts and identify existing knowledge gaps.

## Conservation issues

Concerns over the conservation and sustainable management of large oceanic fish are well established and the Big-eye and Yellowfin tunas identified in this study are listed as vulnerable and near-threatened by the International Union for Conservation of Nature and Natural Resources (IUCN) Red List (*IUCN, 2015*). Somewhat surprisingly, given the high conservation concern of Bluefin tuna species with the red listing of many species as endangered or critically endangered (*IUCN, 2015*) and its inclusion as a product to avoid due to sustainability issues in the Good Fish Guide (*MSC, 2013*), this product was listed on the menus of two restaurants. Bluefin tuna is particularly highly valued for its quality and taste. This would also make it an obvious target for economic fraud, with substitution for a lower value tuna species, as was identified in one case. In another instance, a product labelled with the umbrella term of "tuna" was also identified as Bluefin, which given its premium would appear as a missed promotion opportunity. Perhaps, due to the conservation issues around Bluefin tuna selling this meat under higher anonymity may help conceal that the species or individual was caught illegally (*Jacquet & Pauly, 2008*).

Mercury levels have been highlighted as a concern in some species. Species like Skipjack (*Katsuwonus pelamis*) and Yellowfin, often have lower mercury levels than other tuna species, such as Big-eye and Bluefin, and capture location in certain ocean basins can also be related to differing mercury levels (*Lowenstein, Burger & Jeitner, 2010*; *Burger et al., 2014*). Therefore, knowing what tuna species are being served and where they are caught is not only critical to making conservation informed consumer choices, but is also helpful in minimizing the health concerns of mercury exposure (*Khaksar et al., 2015*). This sort of crucial information is not easily accessible for consumers in restaurants, including sushi bars, and oral enquiries for this type of information appear to be unreliable.

Perhaps less well-known to the general public than conservation issues surrounding tuna, is the fact that most eel species are also of very poor conservation status. The European eel (*Anguilla anguilla*) is regarded as critically endangered (*IUCN, 2015*), and made up 62% of the eel products analysed. American (*Anguilla rostrata*) and Japanese (*Anguilla japonica*) eels were also found among the samples, and these are classified as endangered (*IUCN, 2015*). Although 90% of the freshwater eel consumed are farm-raised, they are not bred in captivity in economically relevant numbers (*Mordenti et al., 2014*; *Okamura et al., 2014*), young eels are still collected in the wild, further threatening wild populations (*Okamura et al., 2014*). The critical status of eels might explain why such a high diversity of species (4) is being found among the total of 21 samples analysed in this study. A worrying pattern of exploitation has already been noticed with eels; when one *Anguilla* species or population becomes over-exploited or fisheries restrictions are imposed, the industry moves to the next in order to fulfil demand (*Crook & Nakamura, 2013*). This may explain the occurrence of 'new' species, such as the Giant mottled eel (*Anguilla marmorata*), identified in the UK market for the first time.

## CONCLUSION

This study detected a relatively low percentage of substitution, which could be an indicator that many restaurants have a positive attitude towards labeling accuracy due to heightened

consumer awareness (*Miller, Jessel & Mariani, 2012*; *Mariani et al., 2014*). Even products, such as tuna, that are typically known to exhibit high levels of mislabeling, showed a remarkable level of compliance, corroborating the idea that seafood trade in the EU is addressing issues concerning mislabeling and food authenticity (*Mariani et al., 2015*). Although the substitutions appear infrequent compared to studies in other territories, or those conducted some years ago, improvements can be made to increase the reliability of the market. The legislation on labelling differs between restaurants, fresh sales and deep-frozen fish. For some groups, such as tuna, snapper or eel, the authorized commercial names cover a large number of species, including species with serious conservation and management issues. In such cases, consumers are unable to choose according to sustainability criteria. Additionally, because our study was restricted to seafood sold in a specific type of food service, at the end of a complex supply chain, it is difficult to determine if fraud is occurring at the landing site, during processing, at the wholesale level, at the retail counter or somewhere else along the way (*Cawthorn, Steinman & Witthuhn, 2012*). Therefore, in such a complex landscape, where restaurants may be just as much victims of mislabelling practices as consumers, more interdisciplinary research will be necessary to identify the mechanisms that still pose a threat to a transparent seafood supply chain.

## ACKNOWLEDGEMENTS

We are grateful to the Academic Editor and three reviewers for their comments on earlier versions.

### Funding
This work was funded by the European Union INTERREG Atlantic Area Program ('LabelFish', project 2011-1/163). The UK Department for Environment, Food and Rural Affairs (DEFRA) grant FA0116, the University of Bristol and the University of Salford. The funders had no role in study design, data collection and analysis, decision to publish, or preparation of the manuscript.

### Grant Disclosures
The following grant information was disclosed by the authors:
European Union INTERREG Atlantic Area Program: 2011-1/163.
UK Department for Environment, Food and Rural Affairs (DEFRA): FA0116.
University of Bristol.
University of Salford.

### Competing Interests
The authors declare there are no competing interests.

### Author Contributions
- Sara G. Vandamme conceived and designed the experiments, performed the experiments, analyzed the data, wrote the paper, prepared figures and/or tables, reviewed drafts of the paper.

- Andrew M. Griffiths conceived and designed the experiments, performed the experiments, analyzed the data, contributed reagents/materials/analysis tools, wrote the paper, reviewed drafts of the paper.
- Sasha-Ann Taylor, Jessica A. Towne and Mhairi Watson performed the experiments.
- Cristina Di Muri prepared figures and/or tables, reviewed drafts of the paper.
- Elizabeth A. Hankard performed the experiments, analyzed the data.
- Stefano Mariani conceived and designed the experiments, contributed reagents/materials/analysis tools, wrote the paper, reviewed drafts of the paper.

## Animal Ethics

The following information was supplied relating to ethical approvals (i.e., approving body and any reference numbers):

The work presented in this paper did not involve the use of live animals and no animals were killed expressly for this work. In fact, all samples we collected from food for human consumption at restaurants. Therefore, ethical oversight is not required in this case.

## Data Availability

All 118 sequences generated in this study have been made publicly available on Genbank (accession numbers KU168615–KU168732) and also appear in Table 1.

## Supplemental Information

Supplemental information for this article can be found online at http://dx.doi.org/10.7717/peerj.1891#supplemental-information.

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
