# Peer review of "Sushi barcoding in the UK: another kettle of fish"

_PeerJ, doi:10.7717/peerj.1891_

## Round 0.1 · original submission · Major Revisions

All of the reviewers find your study potentially suitable for publication, but they have a number of reservations that would need to be addressed in a revision

1. We agree with Reviewer 1 that the data need to be deposited in a public repository this is also PeerJ's policy (https://peerj.com/about/policies-and-procedures/#data-materials-sharing). There should ideally be a table listing your samples, accession numbers and IDs, including any 'unknown' samples, if applicable (e.g., your table 1).

2. There should be a discussion of sample size, and a power analysis, so that the strength of your interpretations can be assessed. This is particularly true, due to the relatively small sample size. Your results should be presented and interpreted with sampling considerations in mind.

Reviewer 1 ·

Basic reporting

There was no molecular data (raw or processed) accompanying this article, so I was unable to perform a complete review. Although such a option is not available with PeerJ, I suggest a "reject and resubmit" decision. Once data can be made available to reviewers, I will be more than happy to look at the manuscript in more detail.

I suggest the authors read Federhen (2011), and the response by Ratnasingham and Hebert (2011) for a better background on this decision. Data should be accessioned to GenBank as unknown environmental samples, and accession numbers, fasta files, and trace chromatogram files should all be made available to reviewers.

http://onlinelibrary.wiley.com/doi/10.1111/j.1755-0998.2011.03054.x/abstract
http://onlinelibrary.wiley.com/doi/10.1111/j.1755-0998.2011.03067.x/abstract

Experimental design

No comments.

Validity of the findings

No comments.

Additional comments

No comments.

Reviewer 2 ·

Basic reporting

The paper is a survey in UK regarding to mislabelling in the sushi restaurants. The concept of food authenticity is a growing area. In the paper Introduction is too long and could be shorten by an efficient way. Too much similar information about fraud in detail has been presented. The authors can write some short information about the concept of authenticity test and sequencing in the introduction section

Experimental design

The sample size is not clear in "sampling" section. Inclusion criteria for samples are not clear as well.

Validity of the findings

Sample size is not clear in sampling section. Although, in the result section the number "61" has been mentioned. sample size of 61 is not enough to conclude that in UK mislabeling is significantly lower than US. The similar studies in US has been performed with couple of hundreds sample size. There are newly published papers related to the topic that authors have not mentioned in their study:
Unmasking seafood mislabeling in U.S. markets: DNA barcoding as a unique technology for food authentication and quality control; Food Control
Volume 56, October 2015, Pages 71–76

Additional comments

Again, due to low sample size, authors should be careful of stating"This study detected a very low percentage of substitution compared to similar investigation the
US".

·

Basic reporting

Vandameme et al conducted a molecular identification of raw fish commercialized within sushi restaurants in the UK, recovering a low substitution rate of 7%. Although very straightforward, minor improvements in data presentation might be worth conducting:

Introduction
Lines90-91 – Why discuss if the specimen was farmed or wild captured? This is not related to the research or results presented.

Line 122- You mention the use of cytochrome b, but results of this analysis was not shown. Moreover, did you try any different primer sets for COI? Also (lines165-167), what was the similarity of your cytb sequences with blasted sequences?

Lines 221-257 – You mentioned before that Tuna species cannot be identified to the species level, and I agree. However, here you are discussing Tuna identification to the species level and its conservation issues. This is trick, don’t you think?

Line 390 – Tables and Figures
Should be nice if you state within the Identification column what the percentage means. Also, where is the cytb sequence?
Moreover, did you find more than one Thunnus species with 100% similarity in the BOLD database? If yes, it is worth mentioning.

Experimental design

No comments

Validity of the findings

No comments

---

## Round 0.2 · Major Revisions

After the initial review, Reviewer 1 asked to see the raw data and submitted an additional review by email (comments below). The reviewer raises some important questions that need to be addressed, particularly with regard to the sequences not being alignable in a single frame, as would be expected for COI fragments. If there are pseudogenes, this may affect the accuracy of the analysis. This issue must be dealt with. The reviewer makes also a helpful suggestion with regard to species identification, and regulatory differences between countries, which may explain some of the results. I hope you find these comments helpful in preparing a revision.

---- START Reviewer 1 email comments:

[1] Now having access to the sequence data, I can see that it is of very low quality. Most of the COI sequences are not alignable in their current state, and do not represent protein coding nucleotides in a viable reading frame. After I ran the sequences through MAFFT, it looked like a 16S alignment! In other words, they are full of single-base insertions, deletions and stop codons. These errors need to be corrected by comparing each chromatogram with a translated protein of a good COI sequence, and re-editing each contig. This is a basic quality-control step that wasn't carried out correctly. Whether this will have any impact on the results, I am not sure, but it should be corrected.

[2] I don't know why the authors didn't utilize the character-based key of Lowenstein et al. (2009) to identify the tuna to species level, and get this useful information at very little effort. It would really add to the paper to know which tuna species are being sold, especially given that the conservation issues are discussed at length. Obviously, this is not possible until the problems above are addressed.

[3] Is it not somewhat disingenuous (see title and abstract in particular) to compare rates of mislabelling between countries that have different marketing rules for selling seafood? It would be a good idea to be a bit more explicit about this and highlight the differences/similarities between the FDA and DEFRA lists and standardize/clarify the comparison a little bit better. It's also perhaps wrong to extrapolate that the seafood sector is better regulated in the UK than the USA, as based upon only looking at sushi, given that there is a greatly reduced number of species involved compared to other types of fish consumption.

---- END Reviewer comments

Reviewer 1 ·

Basic reporting

I am still not able to access any data from this paper. The authors uploaded their sequences to GenBank, but these are not available online yet, and sometimes GenBank can take months to publish these. I'd still like to review this paper properly, but this requires the authors to provide a fasta file of their sequence data to reviewers.

This may seem like a minor detail, but it isn't. Without this data I am unable to say anything about the quality of this manuscript. If I suspect that the authors have made a mistake in their analysis or interpretation I would like to be able to demonstrate this by re-analyzing their data.

Data accessibility to reviewers is mandated in point number four of the PeerJ instructions to authors (see https://peerj.com/about/author-instructions/#preparing-submission):

"Your reviewers must have your raw data or code to review. Please submit:
-As a link to a repository where the data is accessible.
-Uploaded as a supplemental file.
Generally however, there are very few circumstances in which we can accept a manuscript without raw data."

Experimental design

No Comments.

Validity of the findings

No Comments.

Additional comments

No Comments.

·

Basic reporting

I believe that the authors made a great job improving their manuscript by adding more samples and responding adequately to all reviewers comments and suggestions.

Experimental design

ok

Validity of the findings

ok

---

## Round 0.3 · accepted · Accept

The authors did an admirable job improving the manuscript, in the course of two thorough revisions. The current version of the manuscript is suitable for publication.

Reviewer 1 ·

Basic reporting

See previous reviews.

Experimental design

See previous reviews.

Validity of the findings

See previous reviews

Additional comments

Having re-read the article and read the author’s reply with the assurance that their raw data is now in a publishable state, I can’t see a reason why the manuscript cannot be published.